# Fabrication and Characterization of Novel Food-Grade Bigels Based on Interfacial and Bulk Stabilization

**DOI:** 10.3390/foods12132546

**Published:** 2023-06-29

**Authors:** Jiaxi Li, Junze Han, Yahao Xiao, Ruihua Guo, Xinke Liu, Hong Zhang, Yanlan Bi, Xuebing Xu

**Affiliations:** 1College of Food Science and Technology, Henan University of Technology, Lianhua Road, Zhengzhou 450001, China; lijiaxily@126.com (J.L.); hjzzmn@126.com (J.H.); xiaoyahaoo@126.com (Y.X.); bylzry@126.com (Y.B.); 2Wilmar (Shanghai) Biotechnology Research and Development Center Co., Ltd., 118 Gaodong Road, Pudong New District, Shanghai 200137, China; guoruihua@cn.wilmar-intl.com (R.G.); liuxinke@cn.wilmar-intl.com (X.L.); zhanghongsh@cn.wilmar-intl.com (H.Z.)

**Keywords:** bigel, oleogel, hydrogel, zein, beeswax, starch, nanoparticle

## Abstract

Novel food-grade bigels were fabricated using zein nanoparticles for interfacial stabilization and non-surfactant gelators (beeswax and tapioca) for bulk stabilization. The present study demonstrated the importance of interfacial stability for biphasic gels and sheds light on the roles of the gelation mechanism and the oil/water ratio of a bigel on its microstructure, physical properties, and digestion behaviors. The results indicated that it is not an easy task to realize homogenization and subsequent gelation in beeswax–tapioca biphasic systems, as no amphiphilic components existed. However, applying the binding of zein nanoparticles at the oil–water interface allowed us to produce a homogeneous and stable bigel (oil fraction reach 40%), which exhibited enhanced structural and functional properties. Oleogel structures play a crucial role in determining the deformation response of bigel systems. As the oil content increased, the mechanical strength and elastic properties of bigels were enhanced. In the meantime, clear bigel-type transitions were observed. In addition, the fabricated bigels were shown to be beneficial for delayed digestion, and the lowest degree of lipolysis could be found in bigel with 50% oleogel.

## 1. Introduction

Bigels are novel semi-solid materials that comprise two structured phases: oleogels and hydrogels [1]. This structuration of two phases provides superior characteristics in terms of spreadability, mechanical properties, and thermal response in comparison to corresponding singe gels and make bigels ideal candidates for fat-substitutes, 3D-printing materials, meat, etc. In addition, the semi-solid gel structures can immobilize the surrounding inner phase droplets and thus inhibit droplet aggregation [2], which offers a new solution to the poor stabilities of emulsions and emulsion gels during storage. Due to the combined features of both oleogel and hydrogel, bigels are of great interest, as they can deliver both hydrophilic and lipophilic ingredients individually or simultaneously and protect sensitive components from adverse environmental stress [3]. The presence of a hydrogel phase also provides bigels exceptional swelling property that could facilitate the release of inclusions. Moreover, different morphologies can be prepared by a rational design of formulations and procedures, including the hydrogel-in-oleogel (W/O), bi-continuous, oleogel-in-hydrogel (O/W) types [4]. Even so, the greatest attention has been paid to cosmetic and pharmaceutical developments by bigels, and their utilizations in food are still rare. In this context, many efforts have been made to demonstrate their full potential as fat replacers in recent years.

Conventionally, bigels are obtained by directly mixing mono- or multi-component hydrogel and oleogel [5,6]. Their final properties are reported as function of a oleogel/hydrogel ratio, their synthesis parameters as well as natures, and concentrations of gelators [7,8]. In most cases, additional emulsifiers are not necessary to the stabilization of bigel systems, as both phases would rapidly gelate after homogenizing [8]. Still, the lack of interfacial stability may limit their long-term stability [9]. In addition, this could also lead to severe phase separation during shearing [10]. Note that the majority of bigel formulations relevant to food have taken advantage of amphiphilic gelators. Indeed, these amphiphilic components may not only be responsible for the stabilization of a bulk phase but also play a critical role in emulsification during the homogenization step and the stabilization of the whole system. With this in mind, extensive studies are still needed to elucidate the unclear effects of interfacial stability.

Quite recently, Samui et al. (2021) prepared a novel in situ bigel system using a monoglyceride (MG) oleogel and a gelatin hydrogel, with the addition of a surfactant (lecithin) and a co-surfactant (glycerol) [11]. Additionally, they suggested that the surface-active components would reduce the repulsion force between the two phases and stabilize the interface, offering an easier mixing and higher stabilization [11]. Similarly, Golodnizky et al. (2020) further investigated the effect of a surfactant on bigel properties [9]. Sucrose esters (SEs) with a higher esterification degree exhibited higher hydrophilic–lipophilic balance values and diffusion kinetics, resulting in higher SE concentrations in the interface (meaning lower content in the bulk phase) and producing superior rheological and mechanical properties. These results again emphasized the importance of interfacial stabilization in the mixed polar and non-polar systems.

Even so, against the growing demands for healthy attributes in food, consumers tend to prefer foods with as few non-natural ingredients as possible. In this context, Patel et al. (2015) reported, for the first time, an interesting approach to produce bigels using fumed silica nanoparticles and polymers [12]. The resultant bigels displayed a synergistic enhancement in viscosity and complex modulus as compared to the individual phases. Subsequently, Huang et al. (2017) found that the adsorption of complexes of polymers and nanoparticles at the oil–water interface allowed for a stabilization of bigels far from the de-mixing point [13]. Quite recently, Shakeel et al. (2020) demonstrated a bigel that was more adaptable and better mechanically prepared by combining interfacial (silica nanoparticles) and bulk stabilization [10]. These interesting results opened up new possibilities for the development of bigels with enhanced features. However, all these colloidal studies were not food grade, and knowledge on the construction of an interface and bulk-stabilized edible bigels is still missing.

Zein is a natural by-product of corn production with characteristic water insolubility. It is commonly regraded as good candidate for being a water-resistant film and a stabilizer of biphasic systems with its self-assembly properties [14,15]. In the last few years, zein nanoparticles have been developed for acting as attractive Pickering stabilizers. They require a large amount of energy to desorb once they bind to the oil–water interface, and thus provide superior stability to biphasic systems as compared to conventional emulsifiers.

Based on these considerations, to provide a possibility for developing food-grade Pickering bigels and further verify the importance of interfacial stabilization, two non-surfaces active gelators, beeswax (BW) and tapioca, were used to stabilize the bulk phase, and zein nanoparticles were applied as an interface stabilizer. In this way, a novel particle-stabilized bigel was obtained. Different preparation methods of the bigel were first discussed, and, subsequently, the macro properties (mechanical, rheological, digestive properties, etc.) and microstructures were carefully investigated, with respect to both individual gels. In addition, a high storage stability of the resulting gel was postulated and tested by mechanical strength experiments. The present work will help to further unlock the potential of bigels in the food sector.

## 2. Materials and Methods

### 2.1. Materials

Sunflower oil (SFO) was a gift of Kerry Specialty Fats Ltd. (Shanghai, China). BW and tapioca were purchased from Forest Wax Industry Co., Ltd. (Cangzhou, China) and Liangrun Whole Grain Food Co., Ltd. (Xinxiang, China), respectively. Zein (90% purity) was obtained from Longhua Biotechnology Co., Ltd. (Guangzhou, China), whereas Pepsin (≥250 U/mg), pancreatic (8× USP specification), and bovine bile were all the products of Sigma (Shanghai, China), and DF-15 lipase (175 U/mg) was brought from Amano Enzyme (Nagoya, Japan). All other reagents were of analytical grade.

### 2.2. Fabrication of Zein Nanoparticle-Stabilized Bigel

Zein powders were first accurately weighed (1.5% of total gel weight) and dissolved in excess aqueous ethanol (80%, *v/v*) solution (at least 20:1 by weight). The mass of water in the solution was calculated and accurately weighed (based on the weight of the water in each formulation) in advance. The mixture was fully stirred for 2 h at 1000 rpm to ensure adequate hydration. Then, zein suspension could be obtained by removing the ethanol at 40 °C by using a rotary evaporator [15,16,17].

To prepare the aqueous phase of a bigel, tapioca could be added to the suspension, and the mixture would be heated to 85 °C and stirred for 20 min (500 rpm). The oleogel phase of bigel could be obtained by fully dissolving BW (6%, *w*/*w*) in SFO (85 °C, 20 min, 500 rpm).

Finally, a bigel could be formed by slowly pouring a proportionally hot oleogel phase into a hot water phase with high-speed homogenization at 14,000 rpm for 1 min (Ultra-Turrax, T25, IKA, Staufen, Germany). Gel structures were triggered by rapidly cooling down to 4 °C. All these samples were further stored for 24 h to ensure an intact gel structure before analysis. It should be noted that samples with oil fractions below 25% could not be fully homogenized. In addition, zein will tend to self-assemble as oil phase exceeds 80% during the heating step. Thus, in total, 5 bigels with different oleogel fractions (25%, 40%, 50%, 60%, and 75% *w*/*w*) were successfully developed, and they were named as BG-25, BG-40, BG-50, BG-60, and BG-75, respectively. For comparison, corresponding single gels and three non-zein biphasic samples (at 1:1 oleogel/hydrogel ratio) were also prepared: 6% BW oleogel (6BW), 5% tapioca hydrogel (5ST), 8% tapioca hydrogel (8ST), 10% tapioca hydrogel (10ST), 6% BW-5% tapioca biphasic samples (BT-5), 6% BW-8% tapioca (BT-8), 6% BW-10% tapioca (BT-10).

### 2.3. Microscopical Observations

Micrographs were captured by a digital polarized light microscope (Leica DM2000, Canon, Tokyo, Japan) at 200× magnification. To clearly distinguish different phases, samples were dyed with β-carotene. To this aim, β-carotene (0.1%, *w*/*w*) was first dissolved in SFO at 140 °C and cooled down to 85 °C by a water bath. Then, the hot mixture was mixed with BW at 85 °C for 20 min to obtain the hot oleogel phase, which was subsequently used for bigel production. The resultant samples were gently loaded on a glass slide and then covered by a coverslip. At least 6 times per sample were analyzed.

### 2.4. Electrical Conductivity

An electrical conductivity meter (SevenMulti S40, Mettler Toledo, Shanghai, China) was used to verify the micromorphologies of different samples by determining the electrical conductivities [18].

### 2.5. Rheological Characterization

A rotational rheometer (MCR101, Anton Paar, Shanghai, China) with a Peltier temperature-controlling system was used to measure the rheological properties of the bigels. Preliminary amplitude tests were firstly executed to determine the linear viscoelastic region (LVR) at a strain range of 0.01–10%. Then, the storage module (G′), loss module (G′′), and complex module (G*) were recorded by applying a frequency sweep, ranging 0.01 to 10 Hz. Additionally, dynamic temperature ramp tests at constant strain (0.1%) and frequency (1 Hz) were conducted by cooling the sample from 85–20 °C at a rate of 3 °C/min [19]. Subsequently, the apparent viscosity was then monitored between 0.1 and 200 s^−1^ shear rate sweeps. All the tests were performed in triplicate within LVR (0.1%).

### 2.6. Mechanical Properties

A TA-XT texture analyzer (Stable Microsystems, Godalming, UK) with a cylindrical probe (40 mm diameter) was used to determine the textural properties [20]. In the test, 50 g samples were pre-jellified in a cylindrical container (50 mm diameter) and then underwent a back extrusion test. In the test, the pretest speed, and test speed of the extrusion probe were 2 mm^−1^, and the post-test speed was 10.0 mm s^−1^. The extrusion distance and the trigger force were set to 8 mm and 5 g, respectively. The data were calculated by Exponent Software (v.6.1.16.0).

### 2.7. Thermal Analysis

The thermal properties of samples were assessed using a Mettler differential scanning calorimeter (Greifensee, Switzerland). Samples (about 6–8 mg) were first added into hermetic aluminum pans, and an empty pan was set as reference. In the test, samples were equilibrated at 4 °C for 5 min and then heated to 85 °C at 10 °C /min. The thermal profile was recorded by STARe Software (v.15.00, Mettler-Toledo).

### 2.8. Fourier Transform Infrared Spectroscopy

The possible interactions among zein nanoparticles, BW, and tapioca in network structures of freeze-dried bigels were investigated by Fourier transform infrared (FT-IR) experiments, using a iS10 FT-IR spectrophotometer (Thermo Scientific Nicolet, Waltham, MA, USA) [8]. The FT-IR spectra were scanned from 4000 to 400 cm^−1^, with the spectral resolution of 4 cm^−1^ and 32 scans.

### 2.9. Swelling Properties 

The swelling ratios of the bigels were investigated by immersing about 2 g samples (*M*_1_) into deionized water (20 mL) for 10 h [7]. After swelling, the surface water was carefully removed, and the final masses of the samples were recorded as *M*_2_. The swelling ratios could be formulated as:(1)$$Swellingratio(\%)=\frac{M2-M1}{M1}\times 100$$

### 2.10. Simulated In Vitro Digestion 

Two-stage in vitro digestion experiments (pH-STAT) were carried out to evaluate the digestibility properties of the bigesl and their potentials in DHA releases [7,19]. Formulations of the digestive juices are shown in Table 1. For the gastric stage, 20 mL simulated gastric fluids (SGFs) were first mixed with the sample (about 1–2 g) in a 90 mL thermostatic jacketed bottle after pre-heating to 37 °C, and the pH was rapidly adjusted to 2. The simulated gastric lipolysis went for 1 h with constant shaking (100 rpm). For the intestinal stage, immediately after gastric digestion, samples with mixed with 40 mL preheated simulated intestinal fluid (SIF), and the pH of the mixture was then adjusted to pH 6.8 with 0.1 M HCl. The intestinal lipolysis lasted for 2 h.

Lipolysis rates of bigels were calculated using Equation (2) [21]:(2)$$lipolysisrate\left(\%\right)=\frac{V_{NaOH}\times C_{NaOH}\times MMeq}{3\times W_{oil}}$$
where *V_NaOH_* referred to the consumed volume (L) of NaOH in titration process, and *C_NaOH_* was the molarity of NaOH solution (0.25 mol L^−1^). *MMeq* was the average TAG molar mass of SFO, and *W_oil_* was the weight of the digestive samples.

### 2.11. Particle Size and ζ-Potential Measurement

During the digestion process, digesta (20 μL) was collected every 0.5 h and diluted immediately in 10 mL deionized water (0 °C) to terminate the lipolysis reaction. The particle sizes and ζ-potentials of the bigels were analyzed with a Zetasizer Nano ZS90 (Malvern, UK) [22].

### 2.12. Statistical Analyses

Each measurement was conducted at least three times. The graphics were performed by software Origin 9.1. To evaluate significant differences, a one-way ANOVA analysis was carried out by SPSS software (v.25.0) with a significance level of 5%.

## 3. Results and Discussion

### 3.1. Effect of Zein on Bigel Preparation

Apparent observations showed that a significant phase separation was observed in BT-5 (Figure 1A). With further increases in gelator contents, BT-8 and BT-10 were an opaque white color with no phase separation occurring. However, it was found that BT-8 could not form a self-supporting structure through an inversion test (Figure 1B). Although a strong enough structure was built in BT-10 (Figure 1A), a distinct grainy texture was observed. These results suggested that the inhomogeneous mixing and mismatch between each phase of the proposed systems occurred in the absence of surface components.

On the other hand, the results of the inversion test showed that the structure of BG25 collapsed immediately after inversion, indicating the failure of gel formation. No flow and deformation were observed for single gels (6BW and 8ST) under gravity as well as for bigels with oleogel fractions that reached 40%, confirming the successful formation of the gel network. These results confirmed that absorbing zein nanoparticles at the interface could contribute to the ideal emulsification and stabilization of a bigel based on non-surfactant gelators. Among these, the 8ST sample was an opaque white color and had a smooth texture; 6BW showed a yellowish appearance and a greasy texture. All structured bigels also presented a yellowish color with a greasy texture, which became shallower as the oleogel fraction increased.

### 3.2. Microstructure Observation

Insight into the interconnected nature of the fine structures in these samples was given by studying their microstructures (Figure 2). A yellow color represented the presence of an oleogel phase. As expected, the tapioca networks were not captured by polarized light microscopy, whereas the needle-like crystals were found in 6BW [19]. At 40% and 50% oleogel fraction levels, the oily phases of the bigels dispersed as clear spherical oil droplets were wrapped up by the hydrogel phase, indicating the formation of O/W type bigels. Compared to BG40, the droplet sizes in BG50 were much smaller and became more evenly distributed. Similarly, Singh et al. (2022) and Lu et al. (2014) already reported that oil droplet sizes of an inner phase decreased as the oil content improved [23,24]. Oleogels might act as an active or an inactive filler, related to its own nature and particle sizes [8,25]. Generally, a small droplet size was considered to be able to provide more sites for the surfactant and contribute to the interactions between the inner phase and the outer phase [24]. No visible droplets (neither oil nor water) were observed in BG60, suggesting the formation of a bi-continuous phase. At 75% oleogel fraction, the background of the micrograph was full of oil and beeswax crystals, and the distinct water droplets confirmed the formation of W/O type bigels. It is worth noting that visible interfaces were observed for both the droplets in the non-continuous samples (BG40, BG50, and BG75) and for the junction zones between the oleogel and the hydrogel in BG60. This appeared to be evidence of the adsorption of zein particles at the oil–water interfaces.

### 3.3. Electrical Conductivity Test

The electrical conductivity of bigels was used as an indication of which type it would form [18]. The results of the conductivity test fit well with the microstructure observations. In more detail, the conductivity value of the oleogel was found to be close to zero (0.002 ± 0.001 mS/m) due to its natural insulant feature. On the other hand, a single hydrogel exhibited the highest electrical conductivity (49.333 ± 0.115 mS/m). This happened because the free charges or charged groups derived from the continuous water and starch phase could make the materials conductive [2,18]. Furthermore, as the proportion of water phase increased, the conductivities of BG60 (4.560 ± 0.721 mS/m), BG50 (8.493 ± 0.138 mS/m), and BG40 (14.737 ± 1.405 mS/m) increased gradually. These results were in agreement with previous findings by Ghiasi et al. (2022) [18], which proved the existence of a continuous phase. In addition, BG75 was again confirmed to be the W/O type because of an absence of a conducting phase (0.001 ± 0.000 mS/m).

### 3.4. Back extrusion Test

Mechanical behaviors of a bigel have been reported to be greatly affected by its gel type and oleogel distribution [8,26]. The textural properties of all samples were shown in Table 2. The single oleogel showed higher firmness, consistency, cohesiveness, and index of viscosity values than the single hydrogel. In comparison to the individual oleogel and hydrogel, bigels exhibited intermediate consistencies, cohesive properties, and viscosity index properties. In addition, it was also found that BG40 exhibited an equivalent firmness to the oleogel sample, whereas the rising oleogel fractions showed significant enhancing effects on the firmness properties of the bigels. This implied the important role of oleogel rigid structure in bigel systems and also suggested a synergistic effect in strengthening bigel structures. These results were in agreement with previous studies, which already showed a positive proportional relationship in mechanical properties and oil concentrations in guar gum–monostearate bigels [7,23]. Note that BG75 exhibited properties close to 6BW (except for higher mechanical strength), suggesting that the oil phase played a dominant role in determining the final properties of the bigel. On the other hand, a negative relationship between the gel firmness and droplet size was already reported by Golodnizky et al. (2020) and Kim et al. (1996) [9,27]. This happened because bigger surface areas of smaller oil droplets allowed more interactions with continuing matrices and generated a harder gel [28,29]. Thus, the enhancement in firmness could be also partially attributed to the higher structuration degree inner phase, where small gelled water droplets entrapped within the gel matrix could play the roles of active fillers to support the network structures and reinforce the gel strengths [29,30]. As expected, the decrease in oil droplet size conferred a higher stiffness to BG50 than BG40.

Finally, the firmness values of the bigels were also determined during storage at room temperature (25 °C) for a period of 8 months. Different degrees of solid–liquid separations were found in samples BG40 and BG50, whereas BG60 and BG75 were able to remain stable with no liquid leakage. The firmness values of BG60 and BG75 were 298.019 ± 5.714 g (reduced by 36.56% of initial value) and 362.628 ± 9.288 g (reduced by 22.35%), respectively. In the previous work, Wettlaufer and Flöter (2023) reported that the firmness of a beeswax-based oleogel dropped in excess of 50% after 3 months of storage, for both 4%, 8% and 16% wax addition levels [31]. These findings confirmed our previous assumption about the high stability of the particle-stabilized bigel.

### 3.5. Rheological Characterization

The viscoelastic properties of different single gels and bigels were shown in Figure 3. In both cases, G′ values were larger than G′′ values, irrespective of frequency, and marked a solid-like behavior (results not shown). In addition, frequency sweeps revealed that the G′′/G′ values of all samples were in a range of 0.38–0.50, suggesting the formation of typical weak gels [32]. As compared to individual gels, enhanced solid characteristics (G′) were observed for the bigels, indicating a synergistic effect between different phases. This enhancement could have been related to several factors. Firstly, in a mixed system, the networks of oleogel and hydrogel may interpenetrate, leading to further entanglement and arrested de-mixing of the organic and aqueous phases, which could enhance the overall resistance to deformation [12,33]. However, notably, zein was covering the oil–water interface which, to a certain extent, may inhibit the inter penetrating, and thus result in different deformation responses [12]. On the other hand, driven by the hydrophobic interactions, zein nanoparticle aggregates would assemble at the oil–water interface, increase the friction forces against deformation, and, subsequently, facilitate the formation of a more elastic structure [34]. Moreover, the positive interactions between colloidal particles and gelator chains could also play an important role in this synergistic reaction [35]. In addition, the role of the oleogel’s rigid structure in the overall structure cannot be ignored. The increase in oily fraction would also lead to a tighter packing of a dispersed phase and optimize the spatial configuration, thus enforcing the structurization of bigel [7]. As expected, bigels with higher oil contents (BG60 and BG75) showed much higher G′ values than other samples (Figure 3A). Early evidence suggests that the oleogel fraction usually imparts better solid properties to bigels as compared to hydrogels [4,8,36]. Still, the ascendancy phase in determining bigel properties is seemed to be dependent on the oil/water ratio and the intrinsic properties of single gels [35,37]. As shown in Figure 3A, with increasing frequency, the G′ values of the bigels and the 6BW slightly increased, whereas those of the 8ST were almost constant. All bigel samples performed a solid characteristic very close to pure oleogel rather hydrogel. It can be inferred that oleogel played a dominant role in determining the rheological signature of bigel systems, which may arise from its stiffer structure. In terms of viscosity, strong shear thinning behavior was observed for all formulations, indicating a pseudoplastic nature (Figure 3B). The apparent viscosity of samples was decreased by increasing the oleogel fraction, which could be attributed to the lower inherent viscosity of the oleogel phase compared to the hydrogel phase.

The temperature ramp test showed that the G* values of 8ST remained stable as the temperature increased, which suggested that no gel–sol transition occurred (Figure 3C). The increase in G* values in a high temperature range could be related to the incomplete gelatinization of starch in the production process. However, the G* values of 6BW decreased as the temperature increased, suggesting a temperature sensitivity. A sharp decrease in elastic character (G*) took place at a range of 30 °C and 53.6 °C because of the melting of BW crystals’ networks. At a temperature above 53.6 °C, the sample exhibited a liquid characteristic. All bigels exhibited a melting behavior similar to that of oleogel, and, interestingly, the presence of hydrogel structures delayed the collapse process of the bigel network during heating. Thus, a significant increase in T_gel-sol_ was found in the bigel samples as compared to 6BW, where the values obtained for BG40, BG50, BG60, and BG75 were 68.7 °C, 63.0 °C, 57.3 °C, and 56.0 °C, respectively. In addition, this slowing effect appeared to be related to the hydrogel content in the bigel and became more pronounced when the water phase content increased. Similar results have been reported by Zheng et al. (2020), who found that the temperature of gel–solution transition decreased with the oleogel fraction [8]. They also suggested the outer hydrogel could protect samples from an immediate gel-sol transition after the oleogel melted.

### 3.6. Thermal Analysis

The thermal behaviors of different formulated bigels and their corresponding single gels were studied. As shown in Figure 4, characteristic peaks of hydrogel for all samples ranged from 4 to 11 °C during the heating process. The individual oleogel began to melt at 30.8 °C, and a melting peak was obtained at 40.33 °C. In contrast, it was found that the bigels melted at much lower temperatures (24.06–27.33 °C). However, both the melting peaks and the melting completion temperatures of the bigels shifted to higher temperature ranges (43.17–47.33 °C and 54.83–60.15 °C, respectively), which meant the melting profiles were broadened. Additionally, in spite of the observed splitting peak in BG50, there was a tendency for the peak temperature values to increase with a decreasing oleogel content. Moreover, compared to the temperature sweeps results (Figure 3C), temperature values of T_gel-sol_ were higher than the values of melting completion, confirming an effect of the delaying network collapse arising from the hydrogel structures.

### 3.7. Fourier Transform Infrared Spectroscopy

Aiming at understanding the interaction forces occurring in bigels, the FT-IR spectra (at 4000–400 cm^−1^) of all samples were determined (Figure 5). For the single hydrogel, weak characteristic peaks of the C–O antisymmetric stretching vibrations were observed at 1154, 1082, and 1025 cm^−1^, indicating the gelation of starch. The intensities of these peaks corresponded to the structuration degrees of the starch [38]. Among these, the peak at ~1022 cm^−1^ highlighted the presence of amorphous structures [39]. In addition, the broad band at 3700–3200 cm^−1^ represented the -OH stretching vibrations of tapioca, which indicated the presence of H-bonds [40].

For the single oleogel, the C-H stretching vibrational peaks were clearly evident at 3008, 2924, and 2853 cm^−1^, whereas the typical peaks assigned to the H-bonds were not observed. As previously reported, Li et al. (2022) found that the van der Waals interaction, rather than the H-bonds, was the main driving force to stabilize the BW oleogel network [19].

All bigel samples exhibited spectra close to that of the oleogels, and the weak characteristic peaks of the starch hydrogel seemed to be hidden by the peaks attributed to BW. This again confirmed the dominance of the oleogels’ structures in the bigel systems. In addition, no clear peaks were observed for the bigels with different oleogel fractions in the region of 3200–3700 cm^−1^, evidencing the absence of the peaks attributed to H-bonds [41]. In addition, no significant shifts of the signals for the C-H stretching vibrations were obtained with the increase of hydrogel, indicating a lack of (or weak) van der Waals forces [42]. This lack of interaction forces inevitably led to weak affinities between the oleogel and hydrogel phases, which explained the difficulty in bigel formation via direct mixing, as discussed earlier. These results again emphasized the positive role of zein particles in enhancing the internal connectivities and structural strengths by stabilizing the oil–water interface.

### 3.8. Swelling Properties

The swelling ratios of individual gels and different formulated bigels are shown in Figure 6A. As expected, no swelling behavior was found in the single oleogel after soaking due to its hydrophobic character. In bigels, the continuous oleogel networks who inhibited the penetration of water molecules into the interior of the gel system were responsible for the approximately zero swelling ratio of BG75 (0.10 ± 0.05%) [8]. With the increase in hydrogel content, BG60 and BG50 exhibited similar medium swelling ratios (12.83 ± 2.33% and 10.15 ± 1.57%, respectively), and the highest ratio was obtained for the BG40 (27.36 ± 2.12%). This suggested that the oil/water ratio (and bigel type) greatly affected the swelling behaviors of the bigels. As previously reported, the increase in swelling values could be reasonably associated with the swellable starch gel network, as the hydrogel would absorb water and further entangle its network when immersed into water [43]. However, it should be noted that the hydrogel could only absorbed by a small amount of water in the subsequent submersion process, as most of the network space was occupied during gel preparation, even as an external phase [8]. It should be also taken into consideration that the lowest mechanical strength and the largest particle droplets generally implied a loose network with a larger porosity, which was easily accessed for water molecules. Thus, for the same type of bigels, this swelling behavior seemed to be mainly related to the absorption of water into the gap between the oil and water phases. Accordingly, the highest swelling ratio of BG40 could be associated with its highest hydrogel content and weak network structure with the largest particles. The tighter structure from the small oil droplets and the O/W type may have been responsible for the lowest swelling ratio in BG50. For BG60, the partial continuous oleogel phase appeared to confer hydrophobic properties and led to lower swelling.

### 3.9. Simulated In Vitro Digestion

The digestive behavior of the gel matrix is a key factor to evaluate its potential as a fat substitute and a subsequent inclusion releaser. It has been shown that the interfacial composition and size of fat globules can affect the lipid digestive behaviors. Differences in fat particle sizes exhibit a major effect on fat lipolysis and bioavailability rates during digestion progress [19,21]. Commonly, smaller fat droplet sizes are thought to be beneficial for digestion, as they provide a larger surface area for lipase accessing lipids [44]. Still, when it comes to Pickering systems, the situation could be even more complicated because the interface surrounding the droplet could greatly hinder the access of lipase to fat [45]. As such, the ζ-potential could act as an indicator to reflect the changes in interface compositions, interfacial layers, and properties of fat globules during in vitro digestion [21,46].

All the tested samples were in an unstable state (absolute value ζ-potential less than 30 mv) at the beginning of intestine digestion (Table 3) [47]. Additionally, their particle sizes decreased significantly during the digestion process, especially in the first 30 min (except for BG50, discussed later), which indicated a breakdown of the gel network. As shown in Figure 6B, an exponential release pattern was observed for all analyzed samples, where the lipolysis rate first rapidly increased and, subsequently, reached a plateau with a slow growth until the end of digestion. Among the samples tested, the maximum lipolysis rate was determined for the single oleogel (52.47 ± 1.99%), whereas the presence of hydrogel led to lower lipolysis extents for the bigels, regardless of their types (Figure 6A). This suggested that the gelation mechanism seemed to critically affect gel lipolysis, which could be attributed to the presence of a starch network and zein layer covering the oil digestion sites. In bigels, the digestibility of the samples exhibited a downward trend as the hydrogel fraction increased, where a corresponding decrease in hardness was already discussed in TPA analysis (Table 2). It is also worth noting that BG50 showed a much lower lipolysis extent (19.57 ± 2.95%) in comparison with the other bigel samples.

There are several reasons that might account for the differences in the lipolysis degree among the bigel samples. First of all, the cross-linked network based on the oleogel and hydrogel phases could provide physical barriers that hindered the diffusion of the liquid oil as well as the access of the lipase to the oil [48]. Generally, a stronger gel would lead to a lower level of lipolysis. Meanwhile, void spaces might exist among different phases, where the lipase could directly enter for lipolysis [49]. Thus, samples with higher oleogel contents may have more compact structures, preventing lipase contact with substrate oil directly through void spaces. The stronger and denser gel network structure in BG75 was thought to be more difficult to break down during digestion than that of 6BW, which was beneficial for slowing the lipolysis process [50]. Second, the bigel type greatly affected the bigel’s lipolysis. When oleogel acted as the outer phase (BG75), lipase could attack the oil phase directly. However, BG50 and BG40 had the outer hydrogel phases that could further protect the oleogel particles from lipase, leading to delayed lipolysis. This protective effect of the external hydrogel phase may explain the low lipolysis rate of BG40 and BG50. In addition, one should also note that the swelling of the bigel would facilitate the transfer of the liquid oil into the digestive juices [8]. An interesting phenomenon was that there was a good correspondence between the swelling ratios and the final lipolysis rates of the bigels. This suggested that the swelling ratio may have been a predominant feature in determining the digestibility of a bigel. The lowest lipolysis of BG50 could be, therefore, related to the compact network (small void spaces) with a higher mechanical strength and lower swelling, in comparison with BG40. On the other hand, the results of the particle size showed that the particle sizes of samples exhibited a tendency to increase at the initial stage of the intestine digestion stage with the increase in hydrogel phase content. This may have been related to thick protein interfacial layer and the undigested external phase. In the first 30 min of intestine digestion, the particle sizes of BG40, BG60, BG75, and 6BW were found to be significantly reduced, whereas no significant changes occurred in the particle sizes of BG50. This indicated that the gel structure of the BG50 collapsed most slowly during digestion. As the digestion progressed, the interfaces of the BG40, BG60, BG75, and 6BW gradually became stable (absolute value of ζ-potential greater than 30 mv [51,52]), and their particle sizes reached the minimum in 60 min. Note that subsequent possible slight increases in particle sizes were generally thought to be related to the aggregation of small droplets [22]. However, the margin of decrease in the BG50 particle sizes was significantly smaller at 120 min, compared to other samples (Table 3). At the same time, the absolute value of its ζ-potential was significantly lower than 30 mv (23.9 mv). These suggested a possible ongoing digestion in BG50 at the end of its in vitro digestion experiment [47]. Armed with these findings, it can be concluded that BG50 had a significant effect on delaying lipid digestion.

## 4. Conclusions

In summary, novel food-grade bigels were developed by using zein nanoparticles as the stabilizer of oleogel/hydrogel interfaces and by stabilizing the bulk phases based on a three-dimensional network of non-surface gelators (beeswax and tapioca). Combining the interfacial and bulk stabilization, it allowed for well-defined emulsification and gelation in the systems that were usually difficult to gelate directly. The bigel types, physical properties, swelling behaviors, and lipolysis processes of the obtained bigel systems could be tuned by the oil/water ratios. In comparison to single gelator systems, this combination could lead to synergistic effects on mechanical and rheological properties at high oleogel fractions.

In addition, the proposed system could effectively slow the erosion of its network during digestion, leading to delayed digestion, especially at a 1:1 oil/water ratio. These findings expand the possibilities for making diverse food-grade bigels. The fabricated bigels could potentially be used as an alternative to developing reduced-fat food products and matrixes for controlled-releases of lipophilic active components. Given the critical role of interfacial stabilizations in such systems, a more detailed examination of the preparation methods and properties of zein nanoparticles could improve the regulability of gel properties and further demonstrate their full potentials. In addition, mixing speed during the homogenization step had a significant impact on the final properties of the bigels. However, different results could be found in bigel systems with varying structuring mechanisms. Thus, more efforts are needed to investigate the effects of mixing speed on zein-based Pickering bigels for better performance control.

## Figures and Tables

**Figure 1 foods-12-02546-f001:**
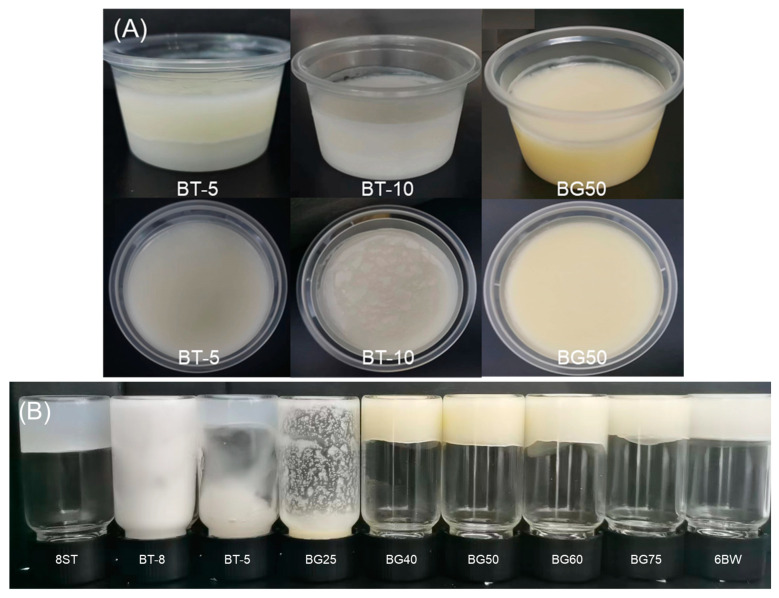
Appearance of biphasic samples with different formulations (**A**); inverted experiment for gelation confirmation (**B**). Abbreviations: BG25, BG40, BG50, BG60, BG75 represent bigels with 25%, 40%, 50%, 60%, 75% oleogel fractions (*w*/*w*), respectively; BT-5 represents biphasic sample with 6% beeswax and 5% tapioca; BT-10 represents biphasic sample with 6% beeswax and 10% tapioca; 6BW oleogel with 6% beeswax; 8ST hydrogel with 8% tapioca.

**Figure 2 foods-12-02546-f002:**
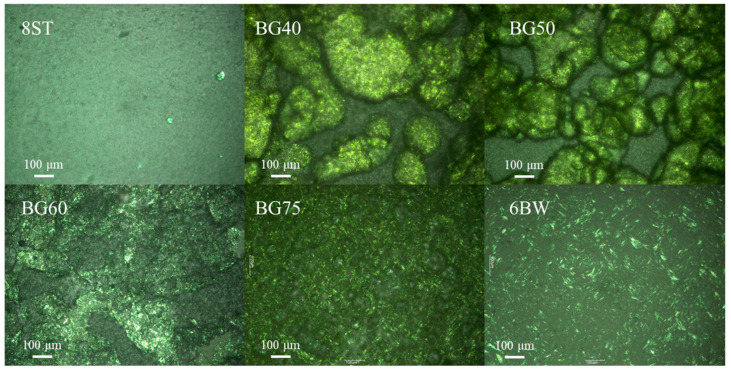
Polarized light microphotographs of single gels and bigels (with different oleogel fractions). Abbreviations: BG25, BG40, BG50, BG60, BG75 represent bigels with 25%, 40%, 50%, 60%, 75% oleogel fractions (*w*/*w*), respectively; 6BW oleogel with 6% beeswax; 8ST hydrogel with 8% tapioca.

**Figure 3 foods-12-02546-f003:**
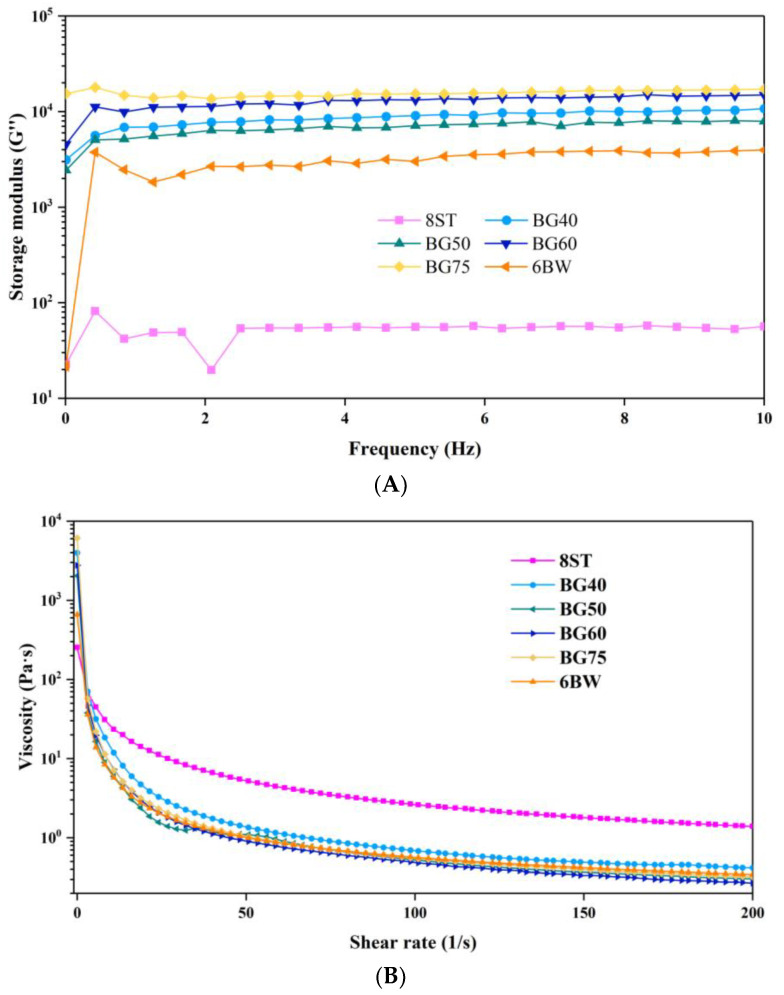
Rheological properties of single gels and different formulated bigels during dynamic shear sweep (**A**), temperature sweep (**B**), and frequency sweep (**C**) tests. Abbreviations: BG40, BG50, BG60, BG75 represent bigels with 40%, 50%, 60%, 75% oleogel fractions (*w*/*w*), respectively; 6BW oleogel with 6% beeswax; 8ST hydrogel with 8% tapioca.

**Figure 4 foods-12-02546-f004:**
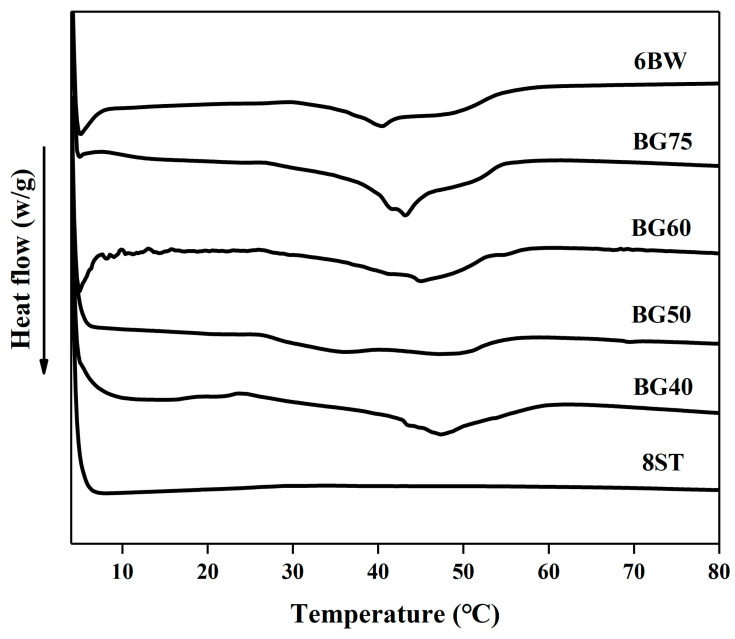
Thermograms of single gels (hydrogel and oleogel) and bigels with different oleogel fractions. Abbreviations: BG40, BG50, BG60, BG75 represent bigels with 40%, 50%, 60%, 75% oleogel fraction (*w*/*w*), respectively; 6BW oleogel with 6% beeswax; 8ST hydrogel with 8% tapioca.

**Figure 5 foods-12-02546-f005:**
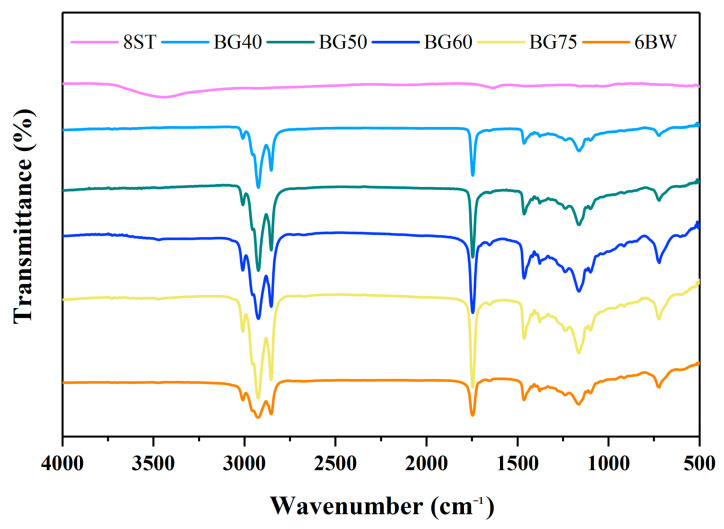
FT-IR spectra of single gels (hydrogel and oleogel) and different formulated bigels. Abbreviations: BG40, BG50, BG60, BG75 represent bigels with 40%, 50%, 60%, 75% oleogel fractions (*w*/*w*), respectively; 6BW oleogel with 6% beeswax; 8ST hydrogel with 8% tapioca.

**Figure 6 foods-12-02546-f006:**
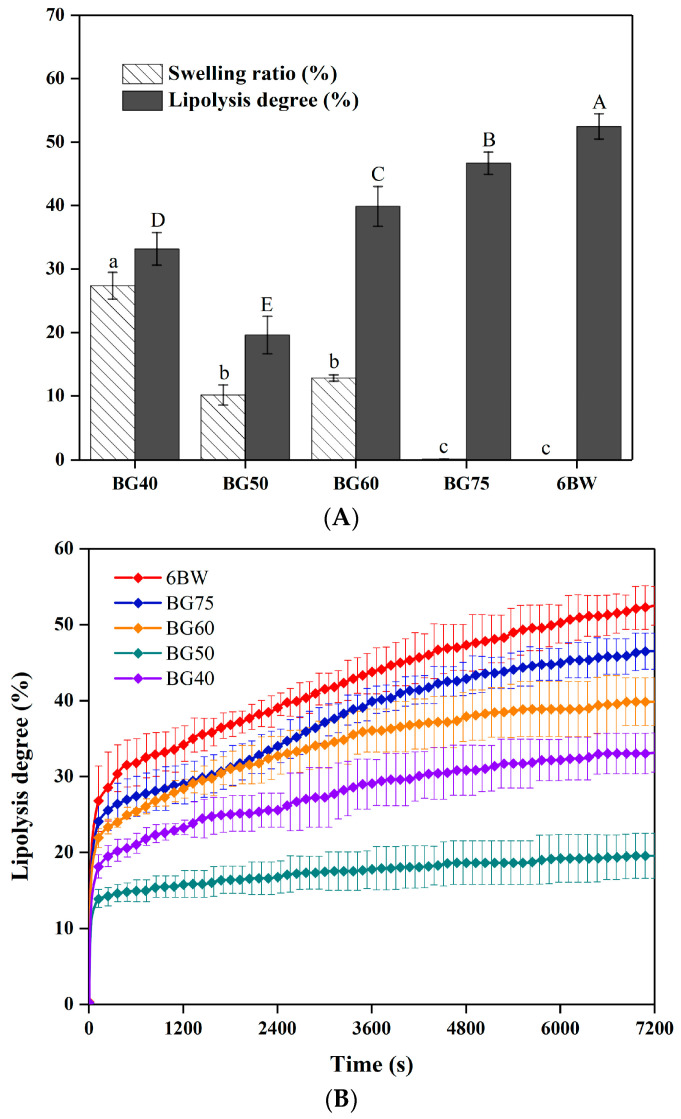
Swelling ratio and lipolysis rate of oleogel and different formulated bigels during the in vitro intestinal lipolysis (**A**); time course curve of in vitro intestinal lipolysis (**B**). Abbreviations: BG40, BG50, BG60, BG75 represent bigels with 40%, 50%, 60%, 75% oleogel fractions (*w*/*w*), respectively; 6BW oleogel with 6% beeswax. Different letters indicate significant differences (*p* < 0.05) among groups (Low case letters are used for swelling ratio results; Capital letters are used for lipolysis degree results).

**Table 1 foods-12-02546-t001:** Composition of digestive juice.

	Compositions	pH
Simulated gastric fluids	NaCl (2.0 mg/mL)	2
	Pepsin (1.28 mg/mL)	
	DF-15 lipase (1.2 mg/mL)	
Simulated intestine fluids	NaCl (8.8 mg/mL)	6.8
	KH2PO4 (6.8 mg/mL)	
	Bovine bile (10 mg/mL)	
	Pancreatic lipase (80 mg/mL)	

**Table 2 foods-12-02546-t002:** Textural properties of single gels and bigels with different oleogel fractions.

Formulations	Hardness (g)	Consistency (g.s)	Cohesiveness (g)	Index of Viscosity (g.s)
6BW	357.930 ± 2.001 ^c^	1351.833 ± 21.349 ^a^	252.493 ± 3.543 ^a^	191.753 ± 6.991 ^a^
BG75	467.013 ± 7.192 ^a^	1372.851 ± 32.115 ^a^	233.926 ± 6.022 ^b^	175.389 ± 8.462 ^b^
BG60	421.047 ± 10.294 ^b^	1288.317 ± 25.341 ^b^	213.931 ± 9.276 ^c^	149.340 ± 5.283 ^c^
BG50	406.988 ± 12.525 ^b^	1130.492 ± 19.920 ^c^	137.810 ± 14.754 ^e^	97.613 ± 9.372 ^d^
BG40	360.912 ± 9.652 ^c^	846.784 ± 11.298 ^d^	165.322 ± 10.094 ^d^	84.660 ± 5.052 ^e^
8ST	79.729 ± 1.020 ^d^	156.006 ± 3.032 ^e^	8.093 ± 0.133 ^f^	3.489 ± 0.219 ^f^

For the same column, a different letter indicates a significant difference in mechanical properties parameters of samples (*p* < 0.05). Abbreviations: BG25, BG40, BG50, BG60, BG75 represent bigels with 25%, 40%, 50%, 60%, 75% oleogel fractions (*w*/*w*), respectively; 6BW oleogel with 6% beeswax; 8ST hydrogel with 8% tapioca.

**Table 3 foods-12-02546-t003:** Particle size (A) and ζ-potential (B) during the in vitro intestinal lipolysis of oleogel and different bigels.

Measurement	Sample	0 min	30 min	60 min	90 min	120 min
Particle size	BG40	1371.3 ± 105.4 ^a^	427.1 ± 16.6 ^cd^	393.7 ± 17.3 ^d^	578.7 ± 54.9 ^b^	514.0 ± 32.0 ^bc^
	BG50	1266.0 ± 52.4 ^a^	1161.0 ± 109.4 ^a^	717.8 ± 57.7 ^b^	767.3 ± 40.7 ^b^	679.8 ± 21.4 ^b^
	BG60	1208.0 ± 39.6 ^a^	675.7 ± 36.5 ^b^	519.3 ± 75.8 ^c^	617.7 ± 33.2 ^b^	483.5 ± 15.9 ^d^
	BG75	1195.0 ± 58.7 ^a^	456.3 ± 20.8 ^b^	375.4 ± 26.1 ^c^	374.2 ± 8.9 ^c^	308.2 ± 3.9 ^d^
	6BW	1011.6 ± 89.0 ^a^	352.8 ± 18.9 ^b^	399.5 ± 17.2 ^b^	355.5 ± 6.3 ^b^	367.0 ± 18.4 ^b^
ζ-potential	BG40	−18.5 ± 0.2 ^a^	−29.3 ± 0.5 ^b^	−37.3 ± 0.6 ^c^	−35.9 ± 2.0 ^c^	−40.1 ± 6.1 ^c^
	BG50	−21.1 ± 0.5 ^a^	−22.9 ± 1.0 ^ab^	−22.8 ± 1.2 ^ab^	−22.9 ± 1.0 ^ab^	−23.9 ± 0.9 ^b^
	BG60	−21.3 ± 2.1 ^a^	−26.3 ± 1.6 ^b^	−28.3 ± 4.4 ^bc^	−26.0 ± 1.1 ^ab^	−31.4 ± 1.2 ^c^
	BG75	−23.2 ± 0.5 ^a^	−31.7 ± 2.3 ^b^	−40.0 ± 6.1 ^c^	−47.3 ± 3.1 ^d^	−44.6 ± 2.5 ^cd^
	6BW	−25.4 ± 1.7 ^a^	−44.8 ± 3.3 ^c^	−41.7 ± 1.2 ^bc^	−43.8 ± 3.1 ^bc^	−40.2 ± 0.9 ^b^

For the same line, a different lowercase letter indicates a significant difference in particle size or ζ-potential (*p* ˂ 0.05). Abbreviations: BG25, BG40, BG50, BG60, BG75 represent bigels with 25%, 40%, 50%, 60%, 75% oleogel fractions (*w*/*w*), respectively; 6BW oleogel with 6% beeswax.

## Data Availability

Data is contained within the article.

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
