# Peer review of "Fabrication and Characterization of Novel Food-Grade Bigels Based on Interfacial and Bulk Stabilization"

_foods, 2023, doi:10.3390/foods12132546_

Round 1
Reviewer 1 Report
In this study novel food-grade bigels are produced using beeswax, tapioca and zein nanoparticles. The topic is of great importance and interesting results are reported. However, the manuscript has several problems.
Comments:
- The language is poor and should be edited by a professional English editor.
- Line 136: In the heading you have written "Texture profile analysis" but in line 139 you have stated that "back extrusion test" was performed. These two tests are completely different and determine different textural parameters. Generally, TPA cannot be performed for oleogel or bigels due to their semisolid nature. TPA is a double compression test which determines parameters such as hardness, adhesiveness, cohesiveness, and gumminess of gels. Whereas, back extrusion is a single compression test for pastes and semi-solid samples. Parameters such as firmness, consistency, index of the viscosity, and cohesiveness are determined in back extrusion. If you have performed back extrusion test please correct the materials and methods and explain each textural parameter. You can use the following reference: 10.1111/ijfs.13222.
- Line 183:You have stated that "The software Origin 9.1" was used for statistical analysis but in line 185 you have written that SPSS software was used. Which software did you use?
- Figure 1: Figure "B" is taken from the surface of this sample. Please change it and use an image similar to "A" and "C".
- Table 2: The statistical analysis is not correct.
-Figure 6: The statistical analysis is not correct.
Table 3: Please explain what is the meaning of different letters. Have you compared different times (0, 30, 60, 90, and120) or different treatments (BG40, 50, 60, 75, and 6BW)?
-Please use "a" for the highest value to prevent mistakes in statistical analysis.
- The results of electrical conductivity test are reported in "microstructure observation". Please write it in a separate section and add more discussion.
- The language is poor and should be edited by a professional English editor.
Author Response
please look at the cover letter for the feedback.

Reviewer 2 Report
The manuscript titled "Fabrication and Characterization of Novel Food-grade Bigels Based on Interfacial and Bulk Stabilization", presents novel results, however the description of the methodology should be improved.
Some observations are as follows
Line 146 Space 5-min
Equation 2, insert a comma after the equation
Line 181 Describe the both techniques, particle size and zeta potential
Author Response
Please look at the cover letter for the feedback.

Reviewer 3 Report
1) in introduction section more clarify the objective and the originalty of this study
2) justify the choice of the different oleogel concentrations?
3) Fabrication of zein nanoparticle, some new references were added in this context.
4) What specific improvements should the authors consider regarding the
methodology? What further controls should be considered?
What specific improvements should the authors consider regarding the
methodology? What further controls should be considered?
Author Response
Please look at the cover letter for the feedback.
Thanks,

Round 2
Reviewer 1 Report
The statistical analysis of Table 2 for the "cohessiveness" parameter in "BG50" and "BG40" is not correct.
The language is fine
Author Response
- Thanks for the comments regarding the statistical analysis of Table 2. It was revised accordingly.
- Are the results clearly presented? The reviewer commented Must be improved. Thanks for the concern. We went through the results and discussion part and re-evaluated the results presentation. According to the comments, we revised the presentation style in each section as much as we can. Hope the presentation has significantly been improved.
- Are the methods clearly described? The reviewer commented can be improved. We gave more details in the method section. The text was also improved in general.
- Is the research design appropriate? The reviewer cmmented can be improved. Thanks for the comment. We evaluated the design and agreed there could be enhanced further for the performance in certain cases, but we also believe the design is sufficient for the study objectives.
